# Shared and Divergent Epigenetic Mechanisms in Cachexia and Sarcopenia

**DOI:** 10.3390/cells11152293

**Published:** 2022-07-25

**Authors:** Laura Yedigaryan, Martina Gatti, Vittoria Marini, Tullia Maraldi, Maurilio Sampaolesi

**Affiliations:** 1Translational Cardiomyology Laboratory, Stem Cell and Developmental Biology, Department of Development and Regeneration, KU Leuven, 3000 Leuven, Belgium; laura.yedigaryan@kuleuven.be (L.Y.); vittoria.marini@kuleuven.be (V.M.); 2Department of Biomedical, Metabolic and Neural Sciences, University of Modena and Reggio Emilia, 41125 Modena, Italy; martina.gatti@unimore.it (M.G.); tullia.maraldi@unimore.it (T.M.); 3Histology and Medical Embryology Unit, Department of Anatomy, Histology, Forensic Medicine and Orthopedics, Sapienza University of Rome, 00185 Rome, Italy

**Keywords:** cachexia, sarcopenia, epigenetics, miRNAs, skeletal muscle, extracellular vesicles

## Abstract

Significant loss of muscle mass may occur in cachexia and sarcopenia, which are major causes of mortality and disability. Cachexia represents a complex multi-organ syndrome associated with cancer and chronic diseases. It is often characterized by body weight loss, inflammation, and muscle and adipose wasting. Progressive muscle loss is also a hallmark of healthy aging, which is emerging worldwide as a main demographic trend. A great challenge for the health care systems is the age-related decline in functionality which threatens the independence and quality of life of elderly people. This biological decline can also be associated with functional muscle loss, known as sarcopenia. Previous studies have shown that microRNAs (miRNAs) play pivotal roles in the development and progression of muscle wasting in both cachexia and sarcopenia. These small non-coding RNAs, often carried in extracellular vesicles, inhibit translation by targeting messenger RNAs, therefore representing potent epigenetic modulators. The molecular mechanisms behind cachexia and sarcopenia, including the expression of specific miRNAs, share common and distinctive trends. The aim of the present review is to compile recent evidence about shared and divergent epigenetic mechanisms, particularly focusing on miRNAs, between cachexia and sarcopenia to understand a facet in the underlying muscle wasting associated with these morbidities and disclose potential therapeutic interventions.

## 1. Introduction

Skeletal muscle represents a highly plastic tissue, with the ability to stretch and contract. Apart from the evident role in movement support and respiration, skeletal muscle functions as a host for a multitude of molecules, as well as a reservoir of heat and protection for internal tissues and organs. Any dysregulation in the skeletal muscle niche is prone to derail a persons’ well-being and overall physical state.

The dysregulation of certain microRNAs (miRNAs) has been implicated in skeletal muscle pathology [1]. miRNAs are small, non-coding RNA sequences involved in regulating gene expression on the post-transcriptional level and have emerged as powerful and important regulators of skeletal muscle development and function. miRNAs bind to target messenger RNAs, leading to either their degradation or translational repression [2]. The role of miRNAs is well-documented in a multitude of biological events, including cell growth, proliferation, differentiation, and cell death [3].

Sarcopenia and cachexia stand for two similar yet very distinct muscle-wasting disorders [4]. Sarcopenia represents muscle wasting associated with aging, whilst cachexia is a secondary disorder to an underlying illness such as cancer. While the role of miRNAs and other epigenetic regulators in these two disorders has been described extensively, the amount of research in this field is increasing due to the stark alteration in the epigenetic landscape of muscle in sarcopenia and cachexia. The present review summarizes the latest research exploring the common and divergent miRNA signatures found in cachexia and sarcopenia, as well as their role in the pathogenesis of these disorders.

## 2. Cachexia

Cachexia is a puzzling syndrome, affecting metabolic processes and related to a causal illness. Cachexia is characterized by a loss of muscle mass that can be accompanied by fat loss, and is often associated with an inflammatory response, insulin resistance, increased protein production, and anorexia [5,6,7,8,9].

The primary outcome of cachexia is severe weight loss. An excess of cytokines has been established as the major cause. Other mediators of weight loss and increased mortality are superfluous myostatin, excess glucocorticoids and testosterone, and insulin-like growth factor-I (IGF-1) deficiency. The prevalence of cachexia is highest in cancer patients, ranging from 40% at diagnosis to 70% in advanced states. Cachexia is also associated with chronic diseases such as chronic heart failure, chronic obstructive pulmonary disease (COPD), chronic kidney disease, and chronic infections [10,11,12,13].

Persistent increase in basal metabolic rate, without the compensation of caloric intake, is the main factor tied to severe weight loss in cachexia. The abnormal metabolic cascade is driven by tumor factors, digestive factors, and hormonal responses to the underlying health issue. Specifically, tumor factors that activate proteolysis and lipolysis contribute to the cachectic phenotype. Digestive factors inducing poor intake of food including nausea, dysphagia, mucositis, dysgeusia, and constipation, inflammatory mediators such as cytokines, and reductions in hormonal mediators such as IGF-1, testosterone, and ghrelin all contribute greatly to cachexia [14]. Mortality is mainly caused by heart failure, instigated by angiotensin-II acting on the ubiquitin-proteasome system, resulting in a significant reduction of protein synthesis and cardiomyocyte apoptosis [15].

Cachexia is also strongly associated with chronic heart failure. Research has found that left ventricular mass correlates with lean body mass in cachectic individuals, signifying that the heart is imperiled to similar consequences to those in lean tissue during cachexia [16].

Currently, the disease stages of cachexia are described as pre-cachexia, cachexia, and refractory cachexia [5,17,18].

Pre-cachexia is defined as a weight loss of more than 1 kg (kg) but less than 5%. Cachexia is defined as a weight loss of more than 5% or when weight loss is more than 2%, with a basal metabolic index (BMI) less than 20 kg/meter (m). Lastly, refractory cachexia is described as weight loss of more than 15% with a BMI less than 23 kg/m, or when weight loss is more than 20% with a BMI of less than 27 kg/m.

Current intervention methods for cachexia include improving appetite and reducing the inflammatory response. The focus of ongoing research is the continuous improvement of outcomes and quality of life. Currently, there is no therapy that contributes to these aspects. The improvement of appetite persists at the center of counteracting cachexia. However, a multimodal approach remains the most effective means of tackling this secondary disease. Interventions that tackle the attenuation of the inflammatory response remain of great importance.

Although counterintuitive, physical exercise can improve cachectic outcomes while aerobic and resistance exercises increase insulin sensitivity and reduce the response to inflammation and oxidative stress. Specifically, for anorexia, the most effective treatment has been megestrol. Megestrol acts by increasing the appetite in patients and is associated with an increased BMI through an increase in body fat. Corticosteroids have also shown a positive effect with regard to appetite, but due to a plethora of side effects, they are only suitable as a short-term intervention. Other medications have also been proven to improve the quality of life of patients. Drugs such as cannabinoids, non-steroidal anti-inflammatory drugs, and omega-3 fatty acids are possible candidates of intervention for cachectic patients and are said to promote improvements in body weight, appetite, and physical functioning among others. Other medications such as β-2 agonists, ghrelin, aliskiren, leucine, and thalidomide are all being studied for their efficacy in improving quality of life. Overall, although many drugs have been used for the management of cachexia, none work reliably or consistently [19,20].

## 3. Sarcopenia

Sarcopenia is a musculoskeletal disorder prominent in elderly populations [21]. It is mainly characterized by a progressive loss of muscle mass. Sarcopenia is defined by the decrease of three specific traits: muscle strength, physical performance, and muscle quality or quantity. These specifications lead to a compromised quality of life and vulnerability to surgical procedures [22,23,24,25,26,27,28].

Sarcopenia is a multifaceted and not entirely understood result of the natural process of aging [29,30,31,32,33,34,35,36]. The development of sarcopenia can be attributed to a multitude of factors associated with aging, such as decreased type II muscle fibers, inactivity, obesity, reduced androgen, and inadequate protein intake. In addition to aging, sarcopenia is associated with and may be caused by several chronic diseases such as chronic obstructive pulmonary disease, chronic kidney disease, chronic heart failure, diabetes mellitus, cancer, and human immunodeficiency virus [37,38,39,40,41,42,43]. Theoretically, these diseases may manifest themselves as primary or secondary effectors e.g., through direct effects on muscle function or through loss of appetite and/or retardation of physical activity [44]. As mentioned for the cachectic cardiac phenotype, the evidence for the revelation of a sarcopenic heart is minimally investigated [38].

The prevalence of sarcopenia ranges between 5–13% in patients aged 60 and above, and 11–50% in patients aged 80 and above [45]. Sarcopenia is almost exclusively a disease affecting elderly populations and does not discriminate based on sex [46].

Several factors remain the main indicators of sarcopenia pathophysiology. These factors include declines in anabolic hormone serum concentrations, neurodegeneration, insulin resistance, and an increase in inflammatory markers [47,48,49,50].

Physical exercise and high protein diets act in a synergistic manner in managing and preventing sarcopenia.

While cachexia and sarcopenia may present largely overlapping symptoms, cachexia is thought to have a more complex etiology compared to sarcopenia. Both diseases may co-exist; however, a patient with a severe chronic disease or cancer is more likely to have cachexia [51]. Furthermore, the Glasgow prognostic score may be utilized to differentiate the two conditions [52]. While cachexia is associated with significant weight loss that leads to a reduction of both fat and fat-free mass, sarcopenia is not outwardly associated with weight loss [53]. Cachexia is also characterized by a more intense inflammatory process as opposed to sarcopenia.

## 4. The Epigenetic Landscape of Cachexia and Sarcopenia in the Context of Myogenesis

Epigenetic modifications play an important role in the development and progression of cancer and subsequent cachexia [54]. Epigenetic modifications describe changes in the condensation state of chromatin, ultimately determining the accessibility of DNA to proteins that control transcription. There are three main epigenetic mechanisms said to play a key role in the development of cancer. These include DNA methylation of CpG islands, post-translational modification of histones, and miRNAs and other noncoding RNAs [55,56,57,58,59,60]. Concerning cancer cachexia, massive evidence has been put forth that epigenetic modifications such as histone acetylation, DNA methylation, and miRNAs orchestrate processes integral to skeletal muscle regeneration and homeostasis (Figure 1).

Epigenetic factors heavily influence the regenerative capacity of satellite cells, adult skeletal muscle cell precursors [61]. Satellite cells are activated by muscle injury, exercise, or disease in which they are induced to proliferate, differentiate, and fuse to form multinucleated myofibers. Factors such as proliferation, differentiation, commitment, and fusion of satellite cells via histone post-translational modifications, miRNAs differential expression, nucleosome positioning, and DNA methylation affect muscle catabolism in cachexia. Satellite cell biology is heavily influenced by histone acetylation. In particular, histone acetyltransferases and deacetylases affect various genes involved in muscle wasting through the modulation of satellite cell activation and differentiation [62]. Studies have shown that Sirtuin 1 (SIRT1) histone deacetylase activity could be reduced by the metabolic switch from fatty acid oxidation to glycolysis during satellite cell activation [63]. This was shown to be responsible for the initiation of muscle gene expression in line with histone 4 lysine 16 (H4K16) acetylation. Other studies demonstrated that the repression of myocyte enhancer factor-2 (Mef2) was resultant from the formation of histone deacetylase 4 (HDAC4)/5-MEF2 complexes, however, upon differentiation, Ca^2+^/calmodulin- dependent protein kinase signaling activation led to the phosphorylation of HDAC and subsequent dissociation of MEF2-HDAC2/5 complexes [64,65,66,67]. HDAC may also complex with retinoblastoma protein (pRb) leading to the interruption of myogenic differentiation 1 (MyoD(1))-HDAC complexes. Overall, at the initiation of Ofer Binah: differentiation, the activity of HDAC may be inhibited through multiple post-translational modifications. The interaction between miRNAs and HDACs also plays an important role in myogenesis, especially during muscle development.

In the skeletal muscle niche, some satellite cells remain quiescent (retaining paired box gene 3 (Pax3) and Pax7 expression) with the ability to self-renew and differentiate [68]. In response to regeneration (differentiation of satellite cells), AKT Serine/Threonine Kinase 1 (AKT1) and AKT2, activated by IGF-1, stimulate the recruitment of p300 acetyltransferase and p300/CBP-associated factor (PCAF) to the specific differentiation gene loci, initiating transcription of myogenic differentiation genes such as MyoD.

Regarding muscle damage caused by cancer-associated muscle catabolism, satellite cells are persistently activated, but fusion is often impaired as a result of unrelenting Pax7 expression [69]. Persistent Pax7 expression prevents proliferating myoblasts from exiting the cell cycle and terminally differentiating to repair damaged myofibers. Similarly, the constant expression of CCAAT/enhancer binding protein (C/EBPβ), a transcription factor that positively regulates atrogin-1 [70], inhibits myogenic cell differentiation. The expression of C/EBPβ is mediated through p300 or HDAC1 interaction and subsequent deacetylation, causing persistent binding of C/EBPβ to the atrogin-1 promoter. In muscles within a wasting environment, C/EBPβ activation blocks the conventional myogenic response. HDAC inhibitors may act by attenuating the regulation of atrogin-1 through C/EBPβ, inhibiting myostatin signaling and initiating AKT signaling.

A crucial cell population for muscle homeostasis is the fibro-adipogenic progenitors (FAPs). These cells contribute to muscle regeneration through the construction of a scaffold that supports muscle repair and regeneration [71]. An HDAC inhibitor, Trichostatin A, can alter the epigenetic landscape of FAPs through the repression of MyoD and follistatin expression in relevant murine models [71,72]. In mice, FAPs have been found to target the SWItch/Sucrose Non-Fermentable (SWI/SNF) chromatin remodeling complex through HDAC activity.

DNA methylation is yet a lesser-known mechanism in the overall cachectic epigenetic landscape. However, mature muscle exudes a unique DNA methylation profile, led by a global DNA demethylation trend [72,73]. Evidence suggests that the intrinsic properties of muscle are ‘programmed’ early in utero or young postnatal life [74,75]. Epigenetic changes, whether transient or stable modifications, may play a role in age-related muscle loss, forwarded by exposure to early-life environmental encounters with regard to skeletal muscle. Adult myogenesis, mainly governed by satellite cells, can be regulated besides DNA methylation by other epigenetic modifications such as chromatin remodeling and histone acetylation, which ultimately affect the expression of myogenic regulatory factors (MRFs), crucial to myogenesis. These epigenetic processes have been described to be impeded with age. Research has shown that satellite cell quiescence and the ability to self-renew are regulated by epigenetic modifications with age [76,77,78,79,80]. Once isolated in vitro, the majority of studies demonstrate that replicative aged satellite cells exude an impaired differentiation phenotype. Satellite cells derived from aged animal models have been shown to have a tendency to become myofibrotic, marked by the overexpression of stem cell antigen-1 (Sca-1) content and loss of MyoD expression [81]. Sca-1 overexpression has been shown to promote the expression of fibrosis-related genes such as proto-oncogene protein (Wnt) and twist-related protein 1 (TWIST1) [82]. It is known that older individuals have a reduced satellite cell pool, in addition to altered myogenic commitment and dysfunctional characteristics. These inadequacies have been associated with global epigenetic modifications. Quiescent satellite cells isolated from both young and aged mice have been assessed epigenetically and compared to activated satellite cells, showing more permissive chromatin states [83]. Activated satellite cells are epigenetically repressed by polycomb-mediated trimethylation (me3) on histone 3 lysine 27 (H3K27). More importantly, aged quiescent satellite cells demonstrate increased trimethylation of lysine 27 on histone H3 (H3K27me3) across the genome compared to young satellite cells. Trimethylation of lysines 9 and 27 on histone H3 (H3K9me3 and H3K27me3) as well as lysine 20 on histone H4 (H4K20me3) are involved in gene suppression [84], suggesting that global gene expression may be suppressed in aged versus young quiescent stem cells by H3K27me3 [83]. It has recently been found that histone demethylase UTX is important for muscle stem cell-mediated regeneration, through the removal of H3K27 trimethylation [85,86]. The study showed that reduced methylation of H3K27 is necessary for muscle regeneration, leading to an increase in myogenin expression and, therefore, differentiation.

Furthermore, DNA methylome array analysis of isolated older human muscle stem cells suggests that these cells possess higher global DNA methylation across the genome [87]. Increased DNA methylation within promotor or enhancer regions typically leads to the suppression of gene expression due to the blocked access of RNA polymerase that enables transcription [88]. Therefore, these reports suggest that disadvantageous histone modifications and genome-wide increases in DNA methylation may further impair gene expression and functionality of older muscle satellite cells.

The active repression of senescence pathways via the polycomb proteins is an important aspect of the maintenance of satellite cell quiescence and self-renewal with age [89]. The polycomb proteins catalyze the acetylation of H3K27 [90] by recruiting histone acetyltransferases such as p300, leading to the activation of gene expression [91]. In addition to this, polycomb proteins may also ubiquitinate histone 2A, making chromatin more accessible to gene expression. Alterations in Sprouty1 (Spry1) have also been associated with muscle stem cell depletion apparent with age [87,92]. Spry1 is an inhibitor of fibroblast growth factor (FGF) and its absence can lead to the activation of FGF2 signaling, resulting in the depletion of the quiescent satellite cell pool. This behavior is typical for aged muscle stem cells. FGF is a known mitogen; therefore, the possible mechanism lies in the activation of muscle cells into S/G2 phases of the cell cycle in lieu of cell-cycle exit [93]. In aged human muscle stem cells, increased DNA methylation of Spry1 has been reported, in parallel with reduced Spry1 transcription [87].

Thus, targeting the epigenome in muscle-wasting conditions is a promising strategy given all that is currently known about the contribution of epigenetics to typical myogenesis.

## 5. A Short Introduction to miRNAs and Their Role in Satellite Cell Regulation

miRNAs are short non-coding RNAs that regulate gene expression on the post-transcriptional level. In mammals, the mechanism of action is either through the binding of miRNAs to target mRNAs (3′ untranslated regions (UTRs)) and sequential mRNA degradation or repression of translation [94] (Figure 2). Most miRNAs are transcribed from DNA sequences into primary miRNAs and further on processed into precursor miRNAs, and finally mature miRNAs. Typically, miRNAs can be secreted into extracellular fluids and shuttled to target cells via extracellular vesicles (EVs) such as exosomes, or by binding to certain proteins. Exosomes are EVs of 30–100 nm that effectively transfer molecules such as proteins, RNA, DNA, lipids, and other molecules to proximal or distal cells [95,96,97]. Exosomes circulate with great stability and protect the enclosed molecules from external degradation.

Highly expressed miRNAs in muscles are represented by the myomiRs [98]. The myomiRs include miR-1, miR-133a, miR-206, miR-208a, miR-208b, miR-133b, miR-499, and miR-486. These miRNAs may be expressed exclusively in skeletal or cardiac muscles, conversely in both types of muscle. The transcription of myomiRs is mainly controlled by myogenic regulatory factors, implicating them in the regulation of skeletal muscle growth and myogenesis and maintenance. MyomiRs have the potential to be used as biomarkers for cachectic and sarcopenic patients, especially following exercise since the myomiRs are strongly regulated during resistance exercise. This can be useful for the avoidance of detrimental exercise [99].

Other than the myomiRs, a multitude of other miRNAs may play a role in both the maintenance of satellite cell quiescence, as well as activation and differentiation [100]. miR-31 has been implicated in the downregulation of the myogenic factor 5 (Myf5) gene in quiescent satellite cells [101]. Another mechanism by which miRNAs regulate satellite cell quiescence is through the targeting of cell cycle regulators. It has been reported that miR-195/497 convert juvenile muscle satellite cells into quiescent ones by targeting factors such as cell division cycle 25C (Cdc25C) and cyclin D (Ccnd) [102]. Consequently, the inhibition of miR-195/497 leads to the reduction of Pax7 expression through the activation of Cdc25C and Ccnd. miR-489 has also been demonstrated to impair satellite cell proliferation through the inhibition of the expression of Dek that regulates mRNA splicing and cell proliferation [103]. Myogenesis has been described to be tightly regulated by miRNAs. As an example, the miR-1/206 family, mir-486, mir-27, and miR-133b have been found to downregulate the expression of Pax3 and Pax7, therefore allowing satellite cell activation [104]. During myoblast C2C12 differentiation, miR-322/424 and -503 have been found to repress the expression of cell division cycle 25A (Cdc25A) [105]. In vivo, miRNAs such as miR-17 and miR-19 have been demonstrated to promote cell differentiation and muscle regeneration after injury [106]. This correlated to the repression of regulators of cell proliferation and the upregulation of myogenic transcription factors myosin heavy chain 3 (Myh3) and MyoD1. miRNAs may also inhibit the process of myogenesis by suppressing MRFs expression. miR-124 overexpression has been associated with low levels of MyoD, Myf5, and myogenin [107]. It has also been demonstrated that in C2C12 cell models, a miR-221/222-MyoD-myomiRs regulatory pathway regulates the expression of some myomiRs such as miR-1, miR-133, and miR-206 [108]. These examples demonstrate the extensive influence that miRNAs have on satellite cells and the overall process of myogenesis.

## 6. Cachexia–Exclusive miRNAs

Although cachexia commonly develops during acute and chronic diseases, it can co-occur with sarcopenia. In fact, these two syndromes overlap considerably, especially in older patients. Unfortunately, the lack of clinical definition, diagnostic criteria, and classification make it difficult to distinguish between cachexia and sarcopenia despite relying on different mechanisms [94,109]. However, cachexia should be distinguished from sarcopenia as it is a multifactorial disease, and different mechanisms underlie the etiology of muscle atrophy in sarcopenia and cachexia. In particular, muscle wasting in cachexia is caused mainly by inflammation [94]. miRNAs play a key role in the onset and development of this pathology, and the ones exclusively involved in cachectic patients are reported in Table 1. Several studies have shown that circulating levels of cytokines are altered in cachectic patients [110]. Recently, changes in metabolism and inflammatory response have been found to be modulated by miRNAs. A study of the plasma from head and neck cancer cachectic patients showed the downregulation of plasmatic miR-130a expression correlated to a higher plasma concentration of tumor necrosis factor-alpha (TNF-α) and risk of being classified as cachectic [111]. The increase in circulating cytokines in cachectic patients elicits an inflammatory response in adipose tissue, which then releases chemoattractant proteins, trapped in a vicious cycle that causes an increase in inflammation and lipolysis [110].

Moreover, it has been found that miR-410-3p was highly expressed in serum exosomes and subcutaneous adipose tissues of cancer-associated cachexia (CAC) patients, which significantly inhibited adipogenesis and lipid accumulation. miR-410-3p targeted insulin receptor substrate 1 (IRS-1) and downstream adipose differentiation factors including C/EBP-α and peroxisome proliferator-activated receptor gamma (PPAR-γ) [112]. In abdominal subcutaneous adipose tissues from cachectic patients with gastrointestinal cancers, miR-99b was downregulated leading to a significant attenuation of adipogenesis targeting stearoyl-CoA desaturase 1 (SCD1) and perilipin 1 (PLIN1) [113].

In a recent study on skeletal muscle tissue from cachectic pancreatic and colorectal cancer patients, six miRNAs were found to be upregulated, namely miR-3184-3p, miR-423-5p, miR-let7d-3p, miR-1296-5p, miR-345-5p, and miR-423-3p [114]. The identified targets of these miRNAs play crucial roles in inflammation, innate immune response, myogenesis, adipogenesis, and signal transduction pathways. It was found that miR-423-5p and miR-3184-3p downregulated genes were involved in lipid biosynthesis. Moreover, the authors showed that let-7d-p targets the transferrin receptor pathway, and its downregulation leads to a reduction in muscle cell proliferation and myogenic differentiation.

Moreover, van de Worp et al. identified 28 significant differentially expressed miRNAs putatively involved in lung cancer cachexia [115]. miRNAs involved in pathways such as interleukin 6 (IL-6), transforming growth factor-beta (TGF-β), TNFα, insulin, and AKT were prominent in cachectic patients. Furthermore, miR-450 was found to be highly expressed in muscles of cachectic patients compared to healthy volunteers. On the other hand, miR-144-5p had lower expression levels as compared to controls.

Moreover, miR-422a has been found to be present in the plasma and muscles of chronic obstructive pulmonary disease (COPD) patients and associated with muscle wasting [113]. In a further study, Paul et al. showed that miR-422a is a suppressor of TGF-β signaling by targeting SMAD Family Member 4 (SMAD4) [116].

Additionally, HuR-RNA binding protein appears to play a critical role in the initiation of cancer cachexia [117]. Through the binding of HuR to the 3′UTR of signal transducer and activator of transcription 3 (STAT3) mRNA, muscle loss appears to be instigated. The region where HuR binds is near the binding region of miR-330. Consequently, the binding of HuR to STAT3 inhibits the binding of miR-330, therefore hindering translational inhibition. Overall, this phenomenon reveals a mechanism of STAT3 modulation and muscle fiber stress response.

Another study evaluated the expression of several miRNAs in the tibialis anterior muscles of C57BL/6J mice with Lewis lung carcinoma. miR-147-3p, miR-299-3p, miR-1933-3p, miR-511-3p, miR-3473d, miR-665-3p, and miR-205-3p were found to be dysregulated. These miRNAs are associated with crucial cellular processes, such as cell-to-cell communication, development, growth, and inflammatory response [118,119]. Notably, miR-205 has been also found to be upregulated in different types of cancer, such as colorectal cancer and nasopharyngeal cancer. miR-205 overexpression is linked to growth suppression, mainly targeting the CAMP responsive element binding protein 1 (CREB1) and phosphatase and tensin homolog (PTEN) [120]. In adult skeletal muscle, miR-205 plays a crucial role in proteasomal degradation, thereby altering muscle proteostasis in a negative manner. Indeed, it was predicted that miR-205 targets the MYC proto-oncogene, bHLH transcription factor (MYC), thus leading to the inhibition of AKT and cascade activation of E3 ligases, atrogin-1/muscle atrophy F-box (MAFbx) and muscle RING finger 1 (MurF-1), the main mediators of muscle protein degradation in cachexia [110,119]. Furthermore, CREB1 is also targeted by miR-450a-5p and was found to be upregulated in skeletal muscles of cachectic non-small-cell lung carcinoma (NSCLC) patients. [115,121].

A study performed on rats with cardiac hypertrophy and failure that develop cachexia showed that miR-132-3p, miR-337-5p, miR-539-5p, miR-136-5p, miR-322-3p, miR-331-3p, miR-386c-3p, miR-204-5p, miR-632, miR-214-3p, miR-489-3p, and miR-30d were differentially expressed in the soleus muscles of the cachectic group compared to the healthy control group. Integrative analyses showed that these miRNAs affect genes involved in extracellular matrix (ECM) organization, respiratory electron transport, and proteasome protein degradation [122].

As stated previously, miRNAs can be transferred in circulation by exosomes [123]. White adipose tissue (WAT) is the major storage space for triacylglycerol (TAG). In cachexia, adipocyte lipolysis is strongly promoted, inducing lipid loss. WAT induces the circulation of inflammatory cytokines, contributing to the initiation of cancer cachexia [124]. Brown adipose tissue (BAT) has been implicated in contributing to the hypermetabolic state of cachexia, however, clinical evidence that BAT contributes to cachexia is limited [125]. WAT can secrete exosomes containing miRNAs that regulate inflammatory processes in immune cells and tissues [28,126]. Di et al. found that cancer-related exosomes are enriched in miR-146b-5p, and the overexpression of this miRNA can result in increased WAT browning, decreased oxygen consumption, and fat mass loss. Further studies identified that miR-146b-5p could directly repress the downstream gene homeodomain-containing gene C10 (HOXC10), thereby regulating lipolysis [127]. Overall, a multitude of miRNAs have been implicated in the pathogenesis of cachexia.

**Table 1 cells-11-02293-t001:** Cachexia—Exclusive miRNAs dysregulated in cachectic patients.

miRNA	Up/Down Regulation	Significant Pathway	Sample	Ref.
miR-let7d-3p	↑	RPS6KA6 PGR CAPN6 SFRP4	Skeletal muscle tissue (rectus abdominus) from cachectic pancreatic and colorectal cancer patients	[113,114,119]
miR-99b	↓	SCD1 Lpin1	Abdominal subcutaneous adipose tissue from cachectic patients	[113]
miR-130a	↓	TNF-α	Plasma of head and neck cancer patients	[111,113]
miR-144-5p	↓	Nrf2	Skeletal muscle of cachectic NSCLC patients	[115,121]
miR-146b-5p	↑	HOXC10	Cancer-related exosomes	[127]
miR-205	↑	CREB1 PTEN	Colorectal cancer cells	[119,120]
	-	BHLH transcription factor (MYC)	Adult skeletal muscle	[119]
miR-345-5p	↑	DLK1 GREM1 CYR61 NOV COL1A1 SOD2 BLNK CAPN6	Skeletal muscle tissue (rectus abdominus) from cachectic pancreatic and colorectal cancer patients	[113,114,119]
miR-410-3p	↑	IRS-1	Exosomes from CAC patients’ serum	[112]
miR-422a	↑	SMAD4	Plasma of COPD patients	[113,116]
miR-423-3p	↑	RET PGR	Skeletal muscle tissue (rectus abdominus) from cachectic pancreatic and colorectal cancer patients	[113,114,119]
miR-423-5p	↑	DLK1 CAMK2A COL1A1 EIF4EBP1 SOD2 CAPN6		
miR-450a-5p	↑	CREB1 AKT/GSK3β	Skeletal muscle of cachectic NSCLC patients	[115,121]
miR-450b *	↑	-	Vastus lateralis of NSCLC patients	[115]
miR-503 *	↑	-	Muscle tissues from ALS-related cachexia patients	[128,129]
miR-542 *	↑	-		
miR-1296-5p	↑	RPS6KA6 PGR CAPN6 SFRP4	Skeletal muscle tissue (rectus abdominus) from cachectic pancreatic and colorectal cancer patients	[113,114,119]
miR-3184-3p	↑	DLK1 GREM1 BMPR1B SQLE RET SOD2 CAPN6 SFRP4		

Abbreviations—COPD: chronic obstructive pulmonary disease, CAC: cancer-associated cachexia, ALS: amyotrophic lateral sclerosis, FFMI: fat-free mass index, NSCLC: non-small-cell lung carcinoma. * miRNA not mentioned in the text.

## 7. Sarcopenia–Exclusive miRNAs

Circulating miRNAs have recently been recognized as novel biomarkers for sarcopenia even though their change in response to sarcopenia is not yet fully understood. In the last few years, different studies tried to demonstrate the roles of specific circulating miRNAs in the onset of this pathology. He et al. analyzed the levels of circulating miRNA levels in the elderly with and without sarcopenia [130]. While miR-1, miR-133a, miR-133b, miR-21, miR-146a, miR-126, miR-221, and miR-20a levels were not changed significantly, plasma miR-208b, miR-222, miR-328d, and miR-499 levels were significantly downregulated in response to sarcopenia. Moreover, they correlated the decreased levels of circulating miRNAs with diagnostic indexes of sarcopenia (ASM/Height2; Handgrip strength and 4-m velocity) in relation to sex.

Furthermore, Ipson and colleagues analyzed the miRNA-exosome content derived from age-related sarcopenic patients [131]. They found that miR-10a-3p, -194-3p, -326, -576-5p, and -760 were increased in sarcopenic patients compared to healthy controls and their expression levels were associated with muscle weight loss, weak grip strength, and self-reported exhaustion.

In another study, Valášková et al. analyzed the expression of myomiRs in blood samples of patients with low muscle performance [132]. Among many, they observed a decrease in the expression of miR-208b and miR-499. These two miRNAs are highly specific skeletal muscle transcripts of the myosin gene. In skeletal muscle, their expression is limited to slow muscle fibers and also plays a key role in muscle fiber displacement and muscle growth promotion [133]. Their downregulation in sarcopenia is linked to the loss of type I muscle fibers in the soleus muscle, reduced expression of slow β-myosin heavy chain, and an increased expression of type IIx/d (fast) myosin isoforms. This finding suggests an impaired intrinsic ability of muscle mass growth in patients with sarcopenia.

Moreover, in an interesting study, Zheng et al. found increased levels of miR-19a and miR-34a in skeletal muscles from sarcopenic patients [134]. Pathway analysis revealed that hsa-miR-34a was significantly enriched in cellular senescence and the mitogen-activated protein kinase (MAPK) signaling pathway. The accumulation of senescent cells and subsequent loss of stem cells may be the linking point between cellular senescence and the aging process [135]. Additionally, the p38 MAPK signaling pathway is implicated in the alteration of satellite cell homeostasis, affecting the autonomous loss and renewal of stem cells [136]. Furthermore, according to Zheng et al., hsa-miR-34a is closely related to the endocrine system and it could be an important player in the aging process. Insulin resistance was found to accelerate muscle protein degradation. In addition to hsa-miR-34a, hsa-miR-19a was another significant node in the miRNA-target gene network. It has been reported that hsa-miR-19a is closely related to the AMP-activated protein kinase (AMPK) signaling pathway, implicated in the control of the aging process by regulating energy metabolism, autophagic degradation, and stress resistance. Loss of AMPK activation increases cell stress and suppresses autophagic clearance, which may contribute to the cell aging process [137].

Adult tissue-specific stem cells contribute to regeneration in response to tissue damage, and the maintenance of satellite cells is essential for the functional homeostasis of skeletal muscle. Crist et al. found that miR-31 is able to control muscle regeneration via regulation of Myf5 [101] highly expressed in quiescent satellite cells. Interestingly, Hughes et al. found that the increase of miR-31 with age contributes to an impaired dystrophin response and increased muscle injury after disuse in old rats (28 months) [138].

An imbalance in protein synthesis and degradation is a major characteristic of muscle wasting in sarcopenia. The most important signaling pathway for muscle protein synthesis is the phosphoinositide 3-kinase/serine-threonine protein kinase/mammalian target of rapamycin (PI3K/AKT/mTOR) signaling pathway. Moreover, different research has highlighted the connection between myogenesis and protein synthesis. For example, it has been reported that miR-128, miR-195, miR-106a, and miR-432 are implicated in the regulatory networks in sarcopenia, targeting the insulin/Ras/MAPK and Phosphoinositide 3-kinase (PI3K)/AKT signaling pathways [139,140,141,142,143,144]. Ma et al. found that miR-432 can negatively regulate myoblast proliferation through the downregulation of E2F transcription factor 3 (E2F3) and phosphoinositide-3-kinase regulatory subunit 3 (P55PIK). miR-432 also suppresses myogenic differentiation by blocking P55PIK-mediated PI3K/AKT/mTOR signaling pathway [143]. Similarly, miR-106a seems to be able to inhibit myogenesis by targeting phosphoinositide-3-kinase regulatory subunit 1 (PI3KR1) [144]. In a recent study, Wang et al., found that miR-487b-3p suppresses the proliferation and differentiation of myoblasts by targeting insulin receptor substrate 1 (IRS1), an essential regulator in the PI3K/AKT and MAPK/extracellular-signal-regulated kinase (ERK) pathways [145]. miR-487b-3p overexpression significantly suppressed C2C12 myoblast proliferation and differentiation, which was accompanied by the downregulation of functional genes related to proliferation (MyoD, Pax7, and proliferating cell nuclear antigen (PCNA)) and differentiation (Myf5, MyoG, and Mef2c).

Human miRNAs described in this section are summarized in Table 2.

## 8. Cachexia–Sarcopenia Shared miRNAs

Sarcopenia and cachexia share some key features such as an increase in oxidative stress and inflammation, an imbalance in muscle protein homeostasis, and muscle cell turnover. In this section, attention will be focused on the main common miRNAs involved in the onset of these pathologies, underlining how the same dysregulated miRNAs could have different implications. Table 3 summarizes the human miRNAs implicated in this section. Several miRNAs have been described to be upregulated in different sarcopenia and cachexia models, such as miR-199a-3p, miR-29b-3p, and miR-27a-3p [114,122,146,147,148]. In skeletal muscle tissue from cachectic pancreatic and colorectal cancer patients, the upregulation of miR-199a-3p was linked to bone morphogenic protein (BMP), ciliary neurotrophic factor (CNTF), IL-8, and mTOR signaling pathway dysregulations and mitochondrial dysfunction [114]. In parallel, Jia et al. observed in an in vitro model of murine myoblast cell sarcopenia that the upregulation of miR-199a-3p partially blocked myogenesis through the suppression of IGF-1/AKT/mTOR signaling pathway [147]. Moreover, it was seen that miR-29b was upregulated in both a rat model of cardiac cachexia and an in vivo mouse model of aging-induced sarcopenia [122,146]. In both cases, the upregulation of this miRNA affected the IGF-1 and PI3K (p85) signaling pathways, leading to an impairment of the TGF-β cellular response, ECM organization, the c-Jun N-terminal kinase (JNK) cascade, and respiratory electron transport. Additionally, the electroporation of miR-29 into the muscles of young mice increased the levels of cellular arrest proteins while in vitro exogenous Wnt-3a stimulated miR-29 expression in a primary culture of muscle precursor cells (MPCs), leading to the suppression of several signaling proteins (p85α, IGF-1, and MYB Proto-Oncogene Like 2 (B-myb)) and consequently, cell proliferation contributing to muscle atrophy [146]. In the rat model of cardiac cachexia, Moraes et al. also observed an increase in miR-27a-5p and McFarlane et al. revealed that tibialis anterior injection with miR-27a antagomirs (sarcopenia in vivo model) leads to skeletal muscle wasting through the reduction of protein synthesis and an increase of ubiquitin-mediated protein degradation [148]. Moreover, a fairly recent meta-analysis regarding the miRNA-mRNA interaction networks associated with cancer cachexia and muscle loss revealed a significant amount of genes differentially expressed in muscle tissues from patients and rodent models of cancer cachexia [149]. It was also revealed that certain miRNA-mRNA interactions may contribute to muscle loss, such as miR-27a/forkhead box protein O1 (Foxo1), miR-27a/Mef2c, miR-27b/C-X-C motif chemokine ligand 12 (Cxcl12), miR-27b/Mef2c, miR-140/Cxcl12, miR-199a/caveolin 1 (Cav1), and miR-199a/Junb.

Otherwise, downregulation of miR-486 has been observed in both sarcopenia and cachexia pathology. In chronic kidney disease (CKD) mouse models, the downregulation of miR-486 was found to be connected to an increase in MAFbx/MuRF1 and atrogin1 expression, and a consequent increase in muscle protein degradation [150]. On the other hand, in a recent study, Liu et al. reported that the downregulation of miR-486 serves as a potential biomarker of sarcopenia-related decline of muscle mass due to the impairment of myoblast differentiation [151]. In fact, in healthy conditions, miR-486 is highly expressed in skeletal muscle and it directly targets Pax7 to promote myoblast differentiation [152,153]. Moreover, it also reduces the expression of PTEN and FoxO1a, in turn phosphorylating AKT and activating the PI3K/AKT pathway [150]. Further experiments showed that in particular, TNFα is a mediator of miR-486 expression in myoblasts [154]. This implies that TNFα, as a mediator of the miRNA circuitry, may be significantly involved in muscle differentiation and survival of cancer cells. Additionally, serum levels of miR-21 have been seen to be increased in both cachectic colorectal cancer patients and sarcopenic post-menopausal women [155,156]. He et al. showed that tumor-derived microvesicles were able to induce apoptosis of skeletal muscle cells via their miR-21 cargo, which targeted the Toll-like 7 receptor (TLR7) [157]. In parallel, Borja-Gonzalez et al. demonstrated that the upregulation of miR-21 in satellite cells and muscle during aging could inhibit myogenesis in vitro via regulation of IL6R, PTEN, and FOXO3 signaling [158]. Moreover, in this study, they demonstrated that inhibition of miR-21 in satellite cells from old mice improved myogenesis. Furthermore, recent studies highlighted the upregulation of miR-424-5p/3p in vastus lateralis muscle samples of COPD and sarcopenic patients [115,159]. In cachectic patients, this increase has been connected to the downregulation of protein synthesis pathways, while in sarcopenic patients, it has been linked to the promotion of atrophy genes such as MuRF-1 and atrogin-1 [160].

As previously described, cachexia is not only characterized by the loss of muscle tissue, a common feature of sarcopenia, but also by a significant increase in adipose wasting. In some cases, the same variation in miRNA expression could, however, have different effects depending on the pathological model in which they are dysregulated. For example, miRNA profiling on abdominal subcutaneous adipose tissue from cachectic patients links the reduction of miR-23a levels to enhanced lipolysis, controlling the expression of transcriptional cofactor peroxisome proliferator-activated receptor-gamma coactivator-1α (PGC1A) [161]. Furthermore, miR-23a regulates glucose transport and interferes with mitochondrial function [162,163]. On the other hand, in sarcopenic rats, Hudson et al. found that atrophy-inducing conditions downregulated miR-23a in muscle by mechanisms involving attenuated calcineurin/nuclear factor of activated T-cells (Cn/NFAT) signaling, and in this condition, leading to an increase of atrogin-1 and MuRF1 expression and subsequently causing atrophy [164]. Moreover, another study found that the injection of leukemia exosomes into mice led to weight and fat loss [165]. In myeloid leukemia cells, K562, as well as K562-derived exosomes, an upregulation of miR-92a-3p was observed. This miRNA can suppress adipogenesis through the downregulation of C/EBPα, implying a role in cachexia mediation. Furthermore, exosomal miRNAs have been documented to induce and maintain cancer cachexia-associated inflammation [110,166]. Conversely, miR-92a overexpression in circulating exosomes of sarcopenic patients is associated with muscle weight loss, weak grip strength, and self-reported exhaustion [131].

Interestingly, miR-483-5p downregulation in cachectic conditions is linked to an increase in lipolysis, targeting IGF2 but its levels were found regulated in an opposite manner in sarcopenic patient’ plasma compared to healthy controls [159]. The most common targets identified were IGF-1 and members of the SMAD family. Functionally, in vitro studies have shown that miR-483-3p inhibits bovine myoblast cell proliferation through the IGF1/PI3K/AKT pathway [167]. Moreover, its upregulation causes a reduction in muscle diameter in mice, and it is also upregulated in muscle-wasting conditions in humans [160]. Furthermore, miR-483-3p may inhibit the TGF-β pathway where known targets include the muscle atrophy genes: atrogin-1 and MuRF-1 [159].

Interestingly, not only the expression of miR-483-3p was seen differentially modulated under sarcopenic or cachectic conditions, but also many other miRNAs, such as miR-532, miR-155, miR-378, and miR-451a [114,115,130,156,163,168,169]. miRNA analysis of abdominal subcutaneous adipose tissue from cachectic patients revealed that the increase in miR-378 levels was linked to an increase in lipolysis, targeting hormone-sensitive lipase (LIPE), patatin-like phospholipase domain-containing protein 2 (PNPLA2), and PLIN1, while in muscle samples of sarcopenic old men, the downregulation was linked to an increase in muscle wasting due to the loss of muscle homeostasis [163,170]. Moreover, elevated expression of miR-378 is also correlated with catecholamine-stimulated lipolysis in adipocytes [113]. Similarly, miR-532 expression was upregulated in skeletal muscle tissue from cachectic pancreatic and colorectal cancer patients but downregulated in muscle and blood samples of sarcopenic aged patients [114,171]. Otherwise, in skeletal muscles of cachectic NSCLC patients, van de Worp et al. saw a reduction in the expression of miR-451a, which is instead increased in the serum of patients with sarcopenia [115,169]. This miRNA is localized on chromosome 17 (17q11.2) and is known to be associated with different human neoplasia, where it is downregulated and contributes to tumor formation and progression [172,173]. miR-451a is expressed in skeletal muscle and in situ, it regulates synaptosomal-associated protein (SNAP-25) and BMP-2 proteins that play an important role in muscle trophism, functionality, and regeneration. Since it also plays a critical role in erythrocyte maturation, it is speculated that increased circulatory levels of miR-451a observed in sarcopenic patients could reflect a compensatory mechanism to induce erythroid maturation, resulting in an increased delivery of oxygen to muscles [169]. Lastly, the expression of miR-155 was found to be upregulated in muscle but downregulated in plasma samples of sarcopenic patients [130,168]. At the same time, Wu et al. demonstrated that breast cancer cell-secreted miR-155 promotes catabolism in resident adipocytes by downregulating the PPAR-γ expression. This may be one of the mechanisms that mediate cancer-induced cachexia [174].

**Table 3 cells-11-02293-t003:** Cachexia/Sarcopenia—shared human miRNAs.

miRNA	Up/Down Regulation	Significant Pathway	Function	Sample	Ref.
miR-21	Cachexia: ↑	TLR7	Increase in myoblast apoptosis	Serum from CRC patients	[155,157]
	Sarcopenia: ↑	IL6 PTEN FOXO3	Decrease of myogenesis	Serum from post-menopausal women (60–85 y)	[156,158]
miR-155	Cachexia: ↑	UCP1 PPARG p-PPARG	Promotion of adipocyte and musclefibre catabolism.Reduction of lipid accumulation	Exosomes from human breast cancer cells (4T1)	[174]
	Sarcopenia: ↓ plasma ↑ muscle	-	-	Sarcopenic patients’ plasma and muscle	[157,168]
miR-378	Cachexia: ↑	LIPE PNPLA2 PLIN1	Increase of lipolysis	Abdominal subcutaneous adipose tissue from cachectic patients with gastrointestinal cancer	[163]
	Sarcopenia: ↓	IGF-1	Loss of muscle homeostasis	Vastus lateralis muscle tissue samples from old (74 ± 2 y) men	[113,170]
miR-424-5p/3p	Cachexia: ↑	UBTF PolR1A RRN3 IGF-1	Reduction of protein synthetic pathway	Vastus lateralis biopsy from COPD patients	[115,160]
	Sarcopenia: ↑	IGF-1 INSR SMAD7	Promotion of TGF-β pathway and atrophy genes (MuRF-1 and Atrogin1)	Vastus lateralis skeletal muscle of adults (>18 y) with sarcopenia	[159]
miR-451a	Cachexia: ↓	-	-	Skeletal muscle of cachectic NSCLC patients	[115]
	Sarcopenia: ↑	SNAP-25 BMP-2	Role in muscle trophism and function	Serum from Caucasian patients with a severe diagnosis of sarcopenia	[169]
miR-483-5p	Cachexia: ↓	IGF2	Enhanced lipolysis	Abdominal subcutaneous adipose tissue from cachectic patients with gastrointestinal cancer/primary human adipocytes	[163]
	Sarcopenia: ↑	IGF1 SMAD family	Inhibition of myoblast cell proliferation through IGF1/AKT/PI3K pathway	Plasma from sarcopenic and obese patients	[159]
miR-532	Cachexia: ↑	GREM1 BMPR1B SULF1 RPS6KA6 NYP1R	BMP signalling; CNTF signalling; Energy metabolisms	Skeletal muscle tissue from cachectic, pancreatic, and colorectal cancer patients	[114]
	Sarcopenia: ↓	BAK1	Increase in apoptosis	Venus’s blood and muscle samples from sarcopenia patients (ages 55–82)	[171]

Abbreviations—CDK: chronic kidney disease, CRC: colorectal cancer, COPD: chronic obstructive pulmonary disease, NSCLC: non-small-cell lung carcinoma, TLR7: toll-like receptor 7, UCP1: uncoupling protein 1, LIPE: hormone-sensitive lipase E, PNPLA2: patatin-like phospholipase domain containing 2, PLIN1: perilipin 1.

## 9. Therapeutic Perspectives—miRNAs and EVs from Stem Cells

Increasing evidence has demonstrated the positive effect of stem cell-secretome alone (as a conditioned medium) in the regeneration of injured tissues [175,176]. Indeed, it is widely accepted that the therapeutic and regenerative potential of stem cells can be largely mediated by paracrine factors such as exosomes [177]. Some in vitro and in vivo recent studies demonstrated the applicability of exosomes derived from stem cells (SCs) for therapeutic purposes. The SC-derived exosomes have demonstrated the potential to treat many diseases and disorders including neurodegenerative diseases [178,179], cardiovascular ischemia [180,181,182], osteoporosis and osteoarthritis [183,184], and kidney and liver diseases [185,186]. EV-based therapy represents a highly potent cell-free therapy approach.

The most interesting findings supporting the regenerative potential of exosomes in cell-free therapy and engineering come from the results obtained from studies on mesenchymal stem cells (MSCs) [187]. In a recent study, Ferguson et al. analyzed the miRNAs-MSC-exosomes content and conducted a network analysis to identify the principal biological processes and pathways modulated by exosomal miRNAs. Here, they observed that miRNA-targeted genes were enriched for cardiovascular and angiogenesis processes, wingless-related integration site (Wnt), TGF-β, and platelet-derived growth factor (PDGF) signaling, proliferation, and apoptosis pathways [188]. Furthermore, Li et al. demonstrated the potential of bone marrow stem cell-derived exosomes (BMSC-exo) to inhibit dexamethasone-induced muscle atrophy in in vitro and in vivo sarcopenia models. They observed that C2C12 myotubes’ miR-486-5p content was upregulated after BMSC-exo treatment and, simultaneously, the nuclear translocation of FoxO1 was downregulated [189]. Likewise, Nakamura and colleagues investigated the paracrine role of MSCs in skeletal muscle regeneration. Interestingly, MSC-exosomes promoted myogenesis and angiogenesis in vitro and muscle regeneration in vivo; the results of this study suggested that muscle regeneration is at least in part mediated by miRNAs such as miR-494 [190].

In the last decade, the efforts of researchers have been directed to the engineering of these vesicles to personalize their content and to use them as delivery systems to target cells or organs [187]. EVs can efficiently deliver RNAs, including miRNAs, which can be eventually loaded via miRNAs’ transduction of EV-releasing cells or transfection of miRNA mimics or precursors [191]. For instance, in a recent study, Wang et al. produced miR-26a-enriched exosomes which exhibited a muscle-specific surface peptide to target muscle delivery. Once injected into the tibialis anterior muscle of cyclin-dependent kinase (CDK) mice, they observed that the overexpression of miR-26a in target regions prevented CDK-induced muscle wasting and attenuated cardiomyopathy via exosome-mediated miR-26a transfer [192].

Other miRNAs have been implicated in the promotion of myogenic differentiation. Among these, the upregulation of miR-26a during myogenesis correlates with a decline in the expression of enhancer of zeste homolog 2 (Ezh2), a chromatin-modifying enzyme that negatively regulates myogenesis [193]. This miRNA has also been demonstrated to inhibit TGF-β signaling and its alteration leads to an inhibition of myoblast differentiation mainly affecting the expression of MyoD and myogenin, a strategy adopted by various miRNAs to regulate myogenesis [152]. Furthermore, there exists a complex network between miRNAs and the epigenetic machinery, wherein the epigenetic regulation of gene expression is strengthened. miRNAs such as miR-29 or the miR-148 families repress the expression of DNA methyltransferases [194,195], while histone modifying enzymes are regulated by miR-101, miR-137, and miR-449a [196,197,198]. MyomiRs such as miR-1, miR-133, and miR-206 regulate the expression of HDACs, either directly or indirectly, creating regulatory circuitries that overlook and maintain the epigenetic regulation of myogenic genes. As an example, miR-1 targets HDAC4 [199], which inhibits the expression of MEF2 [200]. miR-29 targets the transcription regulator Yin Yang 1 (YY1), which, in conjunction with polycomb repressive complex 2 (PRC2) and HDAC1, acts to repress muscle-specific gene expression [201]. HDAC1 controls the expression of miR-206, which, in turn, promotes myoblast differentiation by repressing Pax7 [202].

Exosomal-based therapy may have superiority over cell-based therapies since there is a lack of inherent risk associated with any cell-based therapies, lack of immunogenic response, lack of replicating potential and malignant transformation, and lastly, the potential for a targeted action at the site of interest [203,204]. However, exosome-based therapeutics must first be profiled for safety and follow a standard manufacturing procedure. Nanoparticles and their use as drug carriers are becoming more and more apparent, especially with the manufacturing of some of the coronavirus “COVD-19” vaccinations. However, since exosomes are biological molecules and not synthetically produced, concerns such as the uncertain functioning and intricate composition of exosomes are inquisitive facets that warrant further exploration. If these concerns are answered, the use of these vesicles will represent a highly promising tool in regenerative and personalized medicine (Figure 3).

## 10. Conclusions

There is accumulating evidence that miRNAs are key players in muscle differentiation and are important regulators of muscle atrophy in cachexia and sarcopenia. The overall idea of how miRNA dysregulation alters the specific type of muscle wasting in relation to different pathophysiologies remains uncertain. A myriad of miRNAs is dysregulated in both cachexia and sarcopenia. Some of these miRNA dysregulations are evident in both pathological conditions, however, there is also evidence of exclusive dysregulation and perhaps a promising way of distinguishing these two disorders through these specific molecular signatures. Future studies should focus on the complicated nature of miRNAs and the intricate nature of confounding factors leading to the up- or downregulation of target genes. This may offer an understanding and opportunity to identify key targets for predictive and therapeutic biomarkers.

## Figures and Tables

**Figure 1 cells-11-02293-f001:**
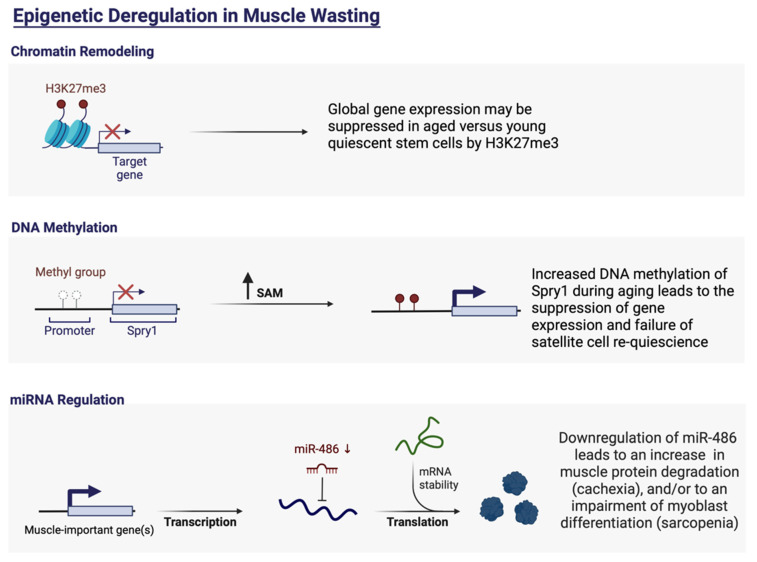
Examples of epigenetic dysregulations in cachexia and sarcopenia. These dysregulations may be due to unfavorable chromatin remodeling, DNA methylation and/or miRNA regulation. Adapted from “Epigenetic Deregulation in Cancer”, by BioRender.com (2022) (accessed on 20 June 2022). Retrieved from https://app.biorender.com/biorender-templates (accessed on 20 June 2022).

**Figure 2 cells-11-02293-f002:**
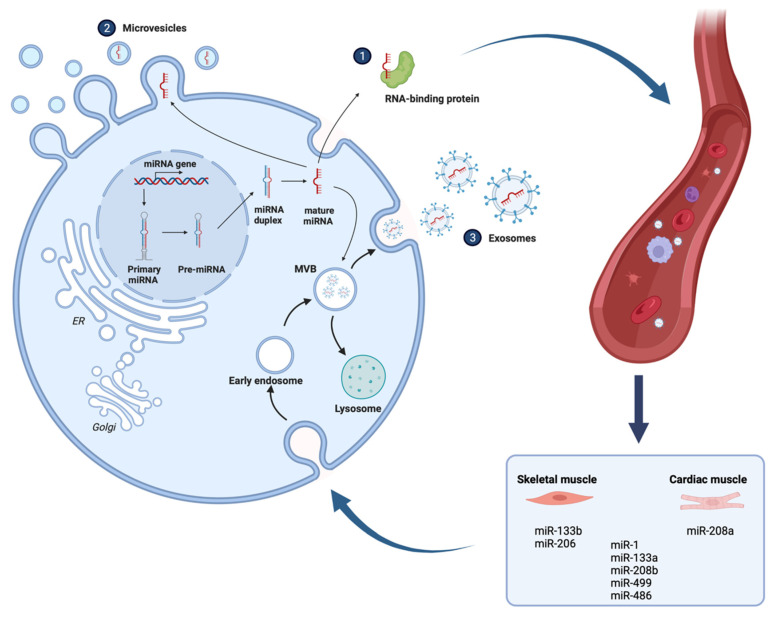
miRNAs can be secreted into extracellular fluids and transported to cells by binding to proteins or by being loaded into microvesicles or exosomes. The eight myomiRs are either expressed in both cardiac and skeletal muscle (miR-1, miR-133a, miR-208b, miR-486, miR-499) or exclusively in one type of muscle (e.g., miR-133b and miR-206 skeletal muscle, miR-208a cardiac). Created with BioRender.com.

**Figure 3 cells-11-02293-f003:**
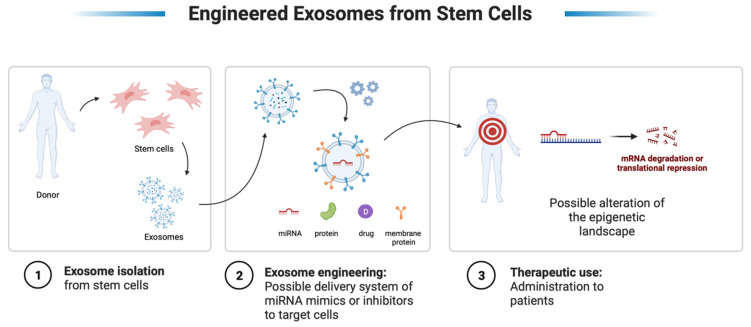
Representation of a possible therapeutic strategy for stem cell-derived exosomes. Created with BioRender.com.

**Table 2 cells-11-02293-t002:** Sarcopenia—Exclusive miRNAs dysregulated in sarcopenic patients.

miRNA	Up/DownRegulation	Significant Pathway	Sample	Ref.
miR-10a-3p	↑	-	Age-related sarcopenic patient exosomes (plasma)	[131]
miR-19a	↑	PRKAA1PFKFB3	Skeletal muscle samples from sarcopenic patients	[134]
miR-34a	↑	MAPK	
miR-194-3p	↑	-	Age-related sarcopenic patient exosomes (plasma)	[131]
miR-208b	↓	-	Plasma samples from older individuals (age ≥ 65 y)	[130]
miR-208b	↓	Myh6Myh7Myh7b	Blood samples of patients with low muscle performance	[132]
miR-222	↓	-	Plasma samples from older individuals (age ≥ 65 y)	[130]
miR-326	↑	-	Age-related sarcopenic patient exosomes (plasma)	[131]
miR-328d	↓	-	Plasma samples from older individuals (age ≥ 65 y)	[130]
miR-499	↓	-
↓	Myh6Myh7Myh7b	Blood samples of patients with low muscle performance	[132]
miR-576-5p	↑	-	Age-related sarcopenic patient exosomes (plasma)	[131]
miR-760	↑	-

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
