# Peer review of "Shared and Divergent Epigenetic Mechanisms in Cachexia and Sarcopenia"

_cells, 2022, doi:10.3390/cells11152293_

Round 1

Reviewer 1 Report

Authors revised literature concerning shared and specific epigenetic signature in cachexia and sarcopenia. The references are mainly recent and the manuscript is very well organized and discussed. For all these reasons the Reviewer consider the mauscript suitable for publication.

Author Response

Authors revised literature concerning shared and specific epigenetic signature in cachexia and sarcopenia. The references are mainly recent and the manuscript is very well organized and discussed. For all these reasons the Reviewer consider the mauscript suitable for publication.

We would like to thank the Reviewer #1 for his/her positive remarks.  We made some changes throughout the review (highlighted in red) to improve readability. We now also provide in the revised ms better tables more usable to the readers and a new  section regarding miRNAs involved in satellite cell  regulations. Finally, we modify the reference format has been changed according to Cells style.

Reviewer 2 Report

In their manuscript, Yedigaryan L. et al. have gathered evidence for the impact of different epigenetic types of machinery on two critical muscle-wasting disorders, Cachexia and Sarcopenia. Although their great focus was given to miRNAs, the authors discussed in depth other epigenetic mechanisms, including DNA methylation and histone acetylation. 

During the last two years, there have been several excellent published reviews tackling the landscape of miRNAs in either cachexia or sarcopenia. However, the authors of this manuscript merged the two disorders and concisely and precisely discussed a large number of different identified miRNAs impacting the diseases. Thus, the manuscript is well written, organized, and detailed. The manuscript offers precise and informative figures and tables.

It would be great if the authors could add a specific section discussing the miRNAs involved in regulating satellite cells.

I suggest that the authors could change the reference style to MDPI.

Author Response

In their manuscript, Yedigaryan L. et al. have gathered evidence for the impact of different epigenetic types of machinery on two critical muscle-wasting disorders, Cachexia and Sarcopenia. Although their great focus was given to miRNAs, the authors discussed in depth other epigenetic mechanisms, including DNA methylation and histone acetylation.

During the last two years, there have been several excellent published reviews tackling the landscape of miRNAs in either cachexia or sarcopenia. However, the authors of this manuscript merged the two disorders and concisely and precisely discussed a large number of different identified miRNAs impacting the diseases. Thus, the manuscript is well written, organized, and detailed. The manuscript offers precise and informative figures and tables.

We would like to thank the reviewer for their wonderfully kind and positive outlook.

It would be great if the authors could add a specific section discussing the miRNAs involved in regulating satellite cells.

We thank the Reviewer#2 for his/her suggestions and  we made some changes throughout the review (highlighted in red) to improve readability. As suggested, we added the section regarding satellite cell regulations to manuscript at the the revised paragraph now entitled: “A short introduction to miRNAs and their role in satellite cell regulations”:

“Other than the myomiRs, a multitude of other miRNAs may play a role in both the maintenance of satellite cell quiescence, as well as activation and differentiation [100]. miR-31 has been implicated in the downregulation of the myogenic factor 5 (Myf5) gene in quiescent satellite cells [101]. Another mechanism by which miRNAs regulate satellite cell quiescence is through the targeting of cell cycle regulators. It has been reported that miR-195/497 convert juvenile muscle satellite cells into quiescent ones by targeting factors such as cell division cycle 25C (Cdc25C) and cyclin D (Ccnd) [102]. Consequently, the inhibition of miR-195/497 leads to the reduction of Pax7 expression through the activation of Cdc25C and Ccnd. miR-489 has also been demonstrated to impair satellite cell proliferation through the inhibition of the expression of Dek that regulates mRNA splicing and cell proliferation [103]. Myogenesis has been described to be tightly regulated by miRNAs. As an example, the miR-1/206 family, mir-486, mir-27, and miR-133b have been found to downregulate the expression of Pax3 and Pax7, therefore allowing satellite cell activation [104]. During myoblast C2C12 differentiation, miR-322/424 and -503 have been found to repress the expression of cell division cycle 25A (Cdc25A) [105]. In vivo, miRNAs such as miR-17 and miR-19 have been demonstrated to promote cell differentiation and muscle regeneration after injury [106]. This correlated to the repression of regulators of cell proliferation and the upregulation of myogenic transcription factors myosin heavy chain 3 (Myh3) and MyoD1. miRNAs may also inhibit the process of myogenesis by suppressing MRFs expression. miR-124 overexpression has been associated with low levels of MyoD, Myf5, and myogenin [107]. It has also been demonstrated that in C2C12 cell models, a miR-221/222-MyoD-myomiRs regulatory pathway regulates the expression of some myomiRs such as miR-1, miR-133, and miR-206 [108]. These examples demonstrate the extensive influence that miRNAs have on satellite cells and the overall process of myogenesis.”

I suggest that the authors could change the reference style to MDPI.

We would like to thank the reviewer for his/her suggestion. The reference style has been changed to match the criteria of MDPI.

Reviewer 3 Report

The review of Yedigaryan et al. describes the different pattern of expression and the diverse function of miRNA in epigenetic mechanisms in cachexia and sarcopenia.

Major cancerns:

-          The review is not easy to read, appears as a long list of miRNAs and is sometimes dispersive. Authors must make the review more usable to a reader. For example, a table showing the differences between miRNAs expressed in sarcopenia and cachexia would help. Even the two tables already present in the paper should be easier for reading and better formatted.

-          Since cachexia and sarcopenia are characterized by metabolic modifications, the authors should insert a paragraph describing how miRNAs affect the metabolic alterations that are present in the two diseases, if known.

-          Line 157: “The epigenetic landscape of cachexia and sarcopenia”. This paragraph does not describe how the epigenetic mechanisms are modified or inducing cachexia or sarcopenia but rather the paragraph describes the epigenetic mechanism during myogenesis. Hence, the title of the paragraph would be changed or the authors would describe the content of the title.  

-           

Author Response

The review of Yedigaryan et al. describes the different pattern of expression and the diverse function of miRNA in epigenetic mechanisms in cachexia and sarcopenia.

Major cancerns:

-          The review is not easy to read, appears as a long list of miRNAs and is sometimes dispersive. Authors must make the review more usable to a reader. For example, a table showing the differences between miRNAs expressed in sarcopenia and cachexia would help. Even the two tables already present in the paper should be easier for reading and better formatted.

We thank the Reviewer #3 for his/her remarks. We agree with Reviewer #3 and we made some adjustments to the review and the tables accordingly.  We made some changes throughout the review (highlighted in red) to improve readability. Since the information reported within the column “Function” of tables was widely described throughout the manuscript, we deleted this column from both tables 1 & 2. Additionally, in order to make the tables more usable to the readers, we arranged the miRNAs included in tables 1, 2 & 3 in ascending order quoting the references.

-          Since cachexia and sarcopenia are characterized by metabolic modifications, the authors should insert a paragraph describing how miRNAs affect the metabolic alterations that are present in the two diseases, if known.

We agree with Reviewer #3 and we thank her/him for picking up this interesting observation. Unfortunately, the information concerning the role of miRNAs specifically affecting the metabolic aspect of the diseases is limited and not exhaustively described in literature to be summarized in an entire paragraph. Nevertheless, we highlighted some miRNAs, such as miR-532, that are involved in the alteration of metabolism related to these complex diseases in the paragraph related to the miRNAs involved in cachexia and sarcopenia.

-          Line 157: “The epigenetic landscape of cachexia and sarcopenia”. This paragraph does not describe how the epigenetic mechanisms are modified or inducing cachexia or sarcopenia but rather the paragraph describes the epigenetic mechanism during myogenesis. Hence, the title of the paragraph would be changed or the authors would describe the content of the title. 

We would like to thank the Reviewer #3  for this remark. We modified the title of this section to “The epigenetic landscape of adult myogenesis.”

Round 2

Reviewer 3 Report

The authors addressed my concerns, hence the paper is suitable for publication.